# Effects of High-Intensity Interval Training on Melatonin Function and Cellular Lymphocyte Apoptosis in Sedentary Middle-Aged Men

**DOI:** 10.3390/medicina59071201

**Published:** 2023-06-26

**Authors:** Hadeel A. Al-Rawaf, Sami A. Gabr, Amir Iqbal, Ahmad H. Alghadir

**Affiliations:** 1Department of Clinical Laboratory Sciences, College of Applied Medical Sciences, King Saud University, Riyadh 11433, Saudi Arabia; hadeelar12345@gmail.com; 2Department of Rehabilitation Sciences, College of Applied Medical Sciences, King Saud University, Riyadh 11433, Saudi Arabia; dr.samigabr@gmail.com (S.A.G.); aalghadir@hotmail.com (A.H.A.)

**Keywords:** melatonin, lymphocyte apoptosis, HIIT training, cytochrome c oxidase

## Abstract

*Background*: Physical performance increased by controlled interventions of high-intensity intermittent training (HIIT); however, little is known about their influence as anti-aging and antioxidant effects, or their role in mitochondrial biogenesis. *Purpose*: This study aimed to determine the effects of HIIT for 12 weeks on melatonin function, lymphocyte cell apoptosis, oxidative stress on aging, and physical performance. *Methods*: Eighty healthy male subjects aged 18–65 years randomly participated in a HIIT-exercise training program for 12 weeks. Anthropometric analysis, cardiovascular fitness, total antioxidant capacity (TAC), lymphocyte count and apoptosis, and serum melatonin and cytochrome c oxidase (COX), were estimated for all subjects before and after HIIT-exercise training. HIIT training was performed in subjects for 12 weeks. *Results*: Data analysis showed a significant increase in the expression levels of the melatonin hormone (11.2 ± 2.3, *p* < 0.001), TAC (48.7 ± 7.1, *p* < 0.002), COX (3.7 ± 0.75, *p* < 0.001), and a higher percentage of lymphocyte apoptosis (5.2 ± 0.31, *p* < 0.003). In addition, there was an improvement in fitness scores (W; 196.5 ± 4.6, VO_2_max; 58.9 ± 2.5, *p* < 0.001), adiposity markers (*p* < 0.001); BMI, WHtR, and glycemic control parameters (*p* < 0.01); FG, HbA1c (%), FI, and serum C-peptide were significantly improved following HIIT intervention. Both melatonin and lymphocyte apoptosis significantly correlated with the studied parameters, especially TAC and COX. Furthermore, the correlation of lymphocyte apoptosis with longer exercise duration was significantly associated with increased serum melatonin following exercise training. This association supports the mechanistic role of melatonin in promoting lymphocyte apoptosis either via the extrinsic mediator pathway or via inhibition of lymphocyte division in the thymus and lymph nodes. Additionally, the correlation between melatonin, lymphocyte apoptosis, TAC, and COX activities significantly supports their role in enhancing physical performance. *Conclusions*: The main findings of this study were that HIIT exercise training for 12 weeks significantly improved adiposity markers, glycemic control parameters, and physical performance of sedentary older adult men. In addition, melatonin secretion, % of lymphocyte apoptosis, COX activities, and TAC as biological aging markers were significantly increased following HIIT exercise training interventions for 12 weeks. The use of HIIT exercise was effective in improving biological aging, which is adequate for supporting chronological age, especially regarding aging problems. However, subsequent studies are required with long-term follow-up to consider HIIT as a modulator for several cardiometabolic health problems in older individuals with obesity.

## 1. Introduction

Human aging is experienced with increased incidences of chronic and serious diseases. Higher rates of heart disease, diabetes, cancer, and declining cognitive capacities such as reduced strength, impaired hearing, and worse memory were reported at the same rate among all individuals with identical chronological ages [1,2]. 

However, a remarkable variation in the rate of biological aging was identified among individuals with identical chronological ages. This, in turn, might be associated with a variation in the incidence of faster age-related declines in health among some adults than others [3,4].

Biological aging refers, in turn, to the gradual and progressive decline in the physiological functions simultaneously involving multiple organ systems within human bodies [5]. Thus, the consequences of differences in genetic endowment, cellular biology, and cellular physiological changes with the accumulation of life experiences across one’s lifespan lead to the divergence of biological age from chronological age among some individuals [6,7,8,9]. 

In addition, acceleration in biological aging is measured by physiological and cellular changes among older adults of the same chronological age. These adults are likely to develop serious diseases such as heart disease, diabetes, and cancer, and have a higher rate of cognitive decline, disability, and mortality [10,11,12].

In midlife, chronological age is adequate to support aging problems, particularly among individuals with rapid aging than other same-age peers [13,14,15]. Thus, the need to study and explore biological aging among older adults or midlife people is required to support and explain aging-related changes [13,14,15].

Biological aging biomarkers have been previously developed and validated along with phenotypic aging as considerable and important markers for identifying the changes that accompany human aging [16,17].

Indeed, understanding the cellular apoptosis of the most important biological systems, such as the immune system and other hormonal changes, as biological aging biomarkers, was required among older adults. Such a way provides more information to build up future models of biological aging.

Apoptosis or cell death was shown as the most important biological mechanism for regulating immune response [18,19]. The killing and elimination of the pathogens needs to induce cell death via apoptosis to activate immune cells, whereas apoptosis acts as a regulatory process to terminate the immunological response [20,21]. Thus, abnormal apoptosis or a missed off-switch regarding apoptosis was significantly reported in chronic inflammation or self-reactive immune responses.

Melatonin (N-acetyl-5 methoxy tryptamine) is a hormone secreted from the pineal gland according to a circadian pattern [22]. Previous studies showed that melatonin could play a role in modulating various biological functions with anti-aging effects [23], anti-inflammatory effects [24], and anti-oxidant activities [25,26]. Furthermore, it protects against chronic diseases, such as cardiovascular diseases, diabetes, and obesity [27,28,29]. Additionally, physically active bodies and practicing exercise are the most significant agents that affect biological aging and protect human bodies from serious diseases among older adults [30,31,32,33].

Physical exercise is very important for maintaining healthy bodies. Physical fitness—including healthy weights, bones, muscles, and joints—physiologically promotes the body against falls and weakness, and reduces surgical risks [34]. Exercise with various intensities and duration was shown to play a role in strengthening the immune system [35]. It was reported that exercise exerts significant hormonal changes, which may affect the mobilization of lymphocytes. Exercise of moderate types was previously shown to increase the secretion of endocrine hormones, lower accumulation of autoreactive immune cells, and subsequently enhance programmed cell death (apoptosis), which reverses the senescence of immune cells. Thus, exercise through physiological hormonal mechanisms can reverse the senescence of immune cells [36].

PA was shown to reduce susceptibility to cancer or degeneration of human cells via increasing melatonin hormonal levels, reducing estrogen production, and improving fat metabolism [37,38]. Additionally, positive apoptotic regulation was observed in various human cell types following participation in moderate exercise training by modulating several physiological factors such as DNA damage, cytokine, reactive oxygen species (ROS), and hormone levels [39].

In metabolically active tissues, the physical activity resulting from varying exercise modes is considered one of the best-known strategies to improve physical performance and overall human health [40,41].

High-intensity interval training (HIIT) as a mode exercise program led to higher improvements than moderate-intensity continuous training (MICT) on functional capacities, muscle function, body composition, and blood biomarkers in obese older adults [42]. In addition, it improves lean mass and skeletal muscle markers of mitochondrial content, fusion, and mitophagy [42].

Thus, as a time-efficient type of training, HIIT could be recommended as an exercise modality to maintain or improve mobility, health, and quality of life among older adults with obesity and other diseases [42,43]. Globally, high-intensity interval training (HIIT) as one type of the traditional exercise modes and other personal and functional fitness training modes scored relatively highly applicable in Europe compared with the United States [44].

However, little is known about the effects of high-intensity interval training (HIIT) on apoptosis regulation mechanism, melatonin function in aging, physical performance, and mitochondrial biogenesis. Thus, in this current study, the effects of melatonin secretion, lymphocyte cell apoptosis, and oxidative stress on aging and physical performance were evaluated in healthy male subjects following HIIT training for 12 weeks.

## 2. Materials and Methods

### 2.1. Subjects

A total of 100 healthy male subjects aged 18–65 years were randomly nominated for this study after having gave written, informed consent. Only 80 subjects who matched the inclusion criteria participated in this study (Table 1 and Figure 1). As the study was performed on healthy subjects, there was a lack of a control group. Instead, the potential effects of HIIT exercise on biological and adiposity markers were estimated from pre-test baseline data and post-HIIT intervention data for 12 weeks. None of the participated subjects had a history of metabolic bone or eating disorders, or chronic or infectious diseases. Additionally, subjects with physical disabilities and who had drugs that could interfere with endocrine hormone secretion, immunity, and antioxidant capacity were excluded from this study. All participants of this study were instructed to have their normal diet during data collection. Anthropometric and body adiposity markers of interest such as body mass index (BMI), hips, waist-to-height ratio (WHtR), conicity index (C-index), and body adiposity index (BAI) were measured using standard measurements before and after training sessions as previously reported [21,22]. This study was approved by the ethical committee of the Ethics Sub-Committee of King Saud University under file number ID: RRC-2017-068.

### 2.2. Exercise Training Program

All subjects were sedentary males identified using a pre-validated global PA questionnaire as previously reported [45,46,47,48]. In that questionnaire, subjects with low activity refers to those with a sedentary lifestyle who have no PA during work and transportation. All subjects participated in a high-intensity interval training (HIIT) program for 40 min/3 sessions/week for 12 weeks using an electronic treadmill (Vegamax, made in Taiwan). The entire HIIT program was performed in the RRC lab, CAMS, King Saud University, and was supervised by an expert physiotherapist with >10 years of experience in his respective field.

The HIIT program consisted of 4 × 4 min intervals at 80–85% of HRmax, with 3 min active recovery at 70% of HRmax between intervals. All subjects started a warm-up for 10 min at 50% of maximal heart rate (HRmax) and went through a 5 min cooldown before initial HIIT program sessions, as previously reported [23]. The heart rate, Borg scale of perceived exertion, as well as the adjustment of the speed of the treadmill were monitored during training to ensure that all subjects were exercising at their corresponding intensity of exercise and to avoid training adaptations during the whole period of the training program. Before and after exercise, venous blood samples (5 mL) were collected from all subjects to estimate melatonin, cytochrome c, lymphocyte count, and apoptosis.

### 2.3. Assessment of Cardiovascular Fitness

A bicycle ergometer (Monark 829E; Ergomedic, Vansbro, Sweden) was used to measure cardiovascular fitness in all subjects. Every third minute, the bicycle was programmed to increase the workload by 50 watts for men and 40 watts for females until exhaustion. A polar heart rate monitor fixed around the chest measured heart rate. In addition, a subjective assessment by a trained observer was performed to determine the exhaustion state, which refers to the participant’s inability to continue training, even with encouragement. To calculate the maximal power output (in Watts [W]), the following formula was used [24], [{W1 + (W2 − W1) × *T/180]}, in which W1 (W) corresponds to workload at the last fully completed stage, W2 (W) corresponds to workload at the final incomplete stage, and T (s) corresponds to time at the last incomplete stage. Cardiovascular fitness was expressed as the maximal output (W) [24].

### 2.4. Assessment of Lymphocyte Count and Apoptosis

Lymphocyte cells were separated from peripheral blood samples (2 mL) to estimate lymphocyte count and viability using trypan blue exclusion and separation tests, as reported previously [25,26,27]. In addition, the DNA-labeling technique was performed to assess lymphocyte apoptosis, which uses acridine orange as a binding fluorescent dye, as mentioned previously in the literature [28].

### 2.5. Assessment of Total Antioxidant Capacity (TAC)

Serum samples of all participants were tested to estimate antioxidant capacity (TAC) as previously reported in the literature [29,30,31]. A colorimetric assay kit (Catalog #K274-100; BioVision Incorporated, CA 95035, USA) was used to perform TAC in this test. The equivalent concentrations of TAC were measured colorimetrically at 570 nm as a function of Trolox concentrations according to the manufacturer’s instructions [29,30,31]. The results reported are calculated according to the following formula;
{[Sa/Sv = nmol/µL or mM Trolox equivalent
where Sa is the sample amount (in nmol) read from the standard curve, and Sv is the undiluted sample volume added to the wells) [29,30,31].

### 2.6. Assessment of Serum Melatonin and Cytochrome C OxidaseX

Immune assay ELISA kits (RE54021. IBL Gesellschaft fur Immunochemie und Immunbiologie MBH, Flughafenstrasse, Hamburg, Germany) were used to estimate serum melatonin in all subjects before and after exercise training. Furthermore, cytochrome c oxidase (COX) was measured in serum samples with competitive ELISA kits obtained from Chemicon International, Temecula, CA, USA. The cytochrome c ELISA kit procedures were performed according to the manufacturer’s instructions [32].

### 2.7. Statistical Analysis

Statistical analysis was performed using SPSS software, version 20.0 (SPSS Inc., Chicago, IL, USA), and results were expressed as mean ± standard deviation (SD). Repeated measures ANOVA was performed to compare changes in the pre- and post-training mean values of BMI, WHtR, melatonin, cytochrome c, TAC, and lymphocyte apoptosis. In addition, the association between serum melatonin and lymphocyte apoptosis (%) with other variables—such as adiposity markers, fitness, TAC, and cytochrome C—were calculated using correlation coefficients (r and β) as tests adjusted for age (42.5 ± 3.1). All data levels obtained at *p* < 0.05 were reported as significant.

## 3. Results

Eighty out of a hundred healthy male subjects aged 18–65 years were recruited in this study. Demographic and clinical characteristics are shown in Table 1. To study the influence of HIIT exercise on the expression levels of melatonin, antioxidant capacity, and related apoptotic markers, the subjects participated in a supervised HIIT exercise program for 12 weeks.

The results showed that adiposity markers, BMI, WHtR, CI-index, and BAI, were significantly decreased (*p* < 0.001) in older adults following HIIT exercise interventions for 12 weeks, as shown in Figure 2A and Table 2. In addition, there was an improvement in fitness scores (W; 196.5 ± 4.6, VO_2_max; 58.9 ± 2.5, *p* < 0.001) and glycemic control parameters (*p* < 0.01); FG, HbA1c (%), FI, and serum C-peptide were significantly improved following HIIT interventions, as shown in Figure 2B and Table 2.

Moreover, the melatonin hormone, TAC, cytochrome c oxidase COX, and lymphocyte apoptosis (LA) were studied as biological aging biomarkers. The data obtained showed that serum levels of the melatonin hormone (11.2 ± 2.3, *p* < 0.001), TAC (48.7 ± 7.1, *p* < 0.002), COX (3.7 ± 0.75, *p* < 0.001), and a higher percentage of lymphocyte apoptosis (5.2 ± 0.31, *p* < 0.003) were significantly increased following HIIT interventions, as shown in Figure 2C,D and Table 2.

Following 12 weeks of the HIIT-exercise program, the exercise duration showed a significant correlation with a higher level of serum melatonin (*p* = 0.001, r = 0.52), and the highest percentage of lymphocyte apoptosis (*p* = 0.001, r = 0.78)) (Table 3). Furthermore, melatonin and lymphocyte apoptosis expression showed a significant correlation with adiposity markers: BMI, WHtR, C-index, BAI, fitness, cytochrome c, and glycemic control variables, as shown in Figure 2A and Table 3. Additionally, lymphocyte apoptosis showed significant associations with the elevation of the melatonin hormone during the progression of the exercise training program (*p* = 0.002, r = 0.37), as shown in Figure 2D and Table 2.

## 4. Discussions

Our study indicated that participating in the HIIT-exercise training program for 12 weeks significantly increases physical performance and fitness, with a significant increase in serum melatonin level as an anti-aging hormone. Furthermore, HIIT-trained subjects showed a significant increase in the levels of TAC, % of lymphocytes apoptosis, and the levels of cytochrome c, and showed significant improvements in adiposity markers, BMI, WHtR, C-index, BAI, and glycemic control variables, especially FG, HbA1c (%), FI, and serum C-peptide. In addition, both melatonin levels and lymphocyte apoptosis as antiaging and physical performance cellular markers showed a significant correlation with exercise duration, fitness, TAC, and a reduction in adiposity markers and glycemic control parameters.

In our study, it seemed that HIIT exercise training for 12 weeks significantly improved adiposity markers (BMI, WHtR, C-index, BAI) and glycemic control variables, especially FG, HbA1c (%), FI, and serum C-peptide.

This means that HIIT may be an effective exercise strategy as an antiadiposity and antidiabetic agent to prevent potential metabolic health impairments among middle-aged people. Several recent studies reported that the interactions between different exercise training modes significantly improve aspects of cardiometabolic health, inflammation, and antioxidant status in inactive, overweight, and obese adults [49,50,51,52,53,54,55,56,57,58,59,60]. It was reported that combination treatments with aerobic and resistance training appear to be more beneficial for improving adiponectin, fasting insulin, fasting blood glucose, insulin resistance index, and triglyceride levels, compared to other exercise modalities [60]. In addition, similar to our results, the effectiveness of aerobic exercise (AE), resistance training (RT), combined aerobic and resistance training (CT), and high intensity interval training (HIIT) on body composition and inflammatory cytokine levels in overweight and obese individuals was recently reviewed using network meta-analysis (NMA). The findings of that review showed that HIIT, along with other combined modes of exercise, could effectively reduce the weight of overweight and obese patients [61,62].

Exercise training with different intensities showed significant changes in the hormonal section of endocrine glands, which play a role in strengthening the immune system and mobilizing lymphocyte cells [6]. Consistent with our data, previous research studies showed a significant increase in daytime melatonin levels in subjects following exercise training [63,64].

The exercise was shown to regulate melatonin secretion via a neural control mechanism, which occurs in controlled consecutive steps: First, there is an increase in the sympathetic discharge, then there is a release of noradrenalin, and finally, it elevates tryptophan levels in pineolocytes. Tryptophan was regarded as the precursor of melatonin synthesis, and its increase in blood circulation increases melatonin synthesis [65,66].

Furthermore, in HIIT-exercise-trained subjects, melatonin secretion significantly correlated with the duration of exercise and fitness scores. It proved that a longer exercise duration might improve melatonin secretion and overall physical performance. These results significantly matched those who reported increased melatonin with longer exercise activity [67]. However, a randomized controlled trial (RCT) showed that a 12-month moderate-intensity exercise intervention did not affect levels of aMT6s in men and women who were previously sedentary. In that study, moderate exercise training did not affect the levels of a urinary metabolite of melatonin 6-sulphatoxymelatonin (aMT6s) in subjects with previously sedentary activities [68].

HIIT and lower IIT exercise programs have effectively increased aerobic and anaerobic capacity in healthy, active men, which may induce metabolic and performance adaptations comparable to traditional endurance training [69]. In addition, HIIT exercise modes may promote an increase in melatonin synthesis and secretion [39,70] via modulating the relationships between melatonin and other hormones such as norepinephrine and epinephrine [39,70].

In this study, HIIT-exercise-trained subjects also showed significant improvements in the levels of total antioxidant capacity (TAC), adiposity markers BMI and WHtR, as well as glycemic control variables FG, HbA1c (%), FI, and serum C-peptide. Previous research showed that modulation and control of significant physiological processes, particularly DNA damage, oxidative stress, and hormonal changes in various cell types, depend mainly on the type of exercise, and its intensity, frequency, and duration as the training endpoint [68,69,70,71]. Furthermore, prolonged exercise training of at least moderate intensity improves TAC, insulin sensitivity, and HbA1c (%) levels, with improved adiposity markers in men with type 2 diabetes [72,73,74,75,76].

In addition, melatonin secretion in subjects showed a significant association with improved TAC, diabetes variables, and adiposity among subjects who participated in 12 weeks of HIIT-exercise training. Several experimental studies showed that supplementation of melatonin plays a significant role in suppressing body weight, visceral adiposity, plasma leptin, and insulin levels [77,78,79,80]. Thus, our data showed that HIIT exercise significantly improves the physical performance of healthy subjects by increasing melatonin expression, which may have potential therapeutic effects against insulin resistance, oxidative stress, and some aging-associated behavioral changes, and that the reduction in melatonin with aging alters metabolism and physical activity, resulting in obesity and associated detrimental metabolic consequences [81].

Physical exercise performance was increased in terms of both aerobic and anaerobic exercise capacities in subjects following high-intensity intermittent exercise (HIIE) training [80,82,83]. HIIT training increased muscle glycolytic and oxidative activities, particularly hexokinase, hydroxyl acyl-CoA dehydrogenase, citrate synthase, the expression of glucose transporter 4 (GLUT4), and peroxisome activated receptors, which collectively contain the master regulator of mitochondrial biogenesis in the skeletal muscles [84,85,86].

This study estimated TAC and COX as markers of physical performance. The data showed a significant increase in TAC and COX levels following HIIT training for 12 weeks. The data agreed with those who reported increased COX biogenesis in mice following moderate exercise of 60 min, 6 days a week [87]. Additionally, subjects with moderate to active physical activity showed a significant increase in TAC, and lower physical activity was associated with poor muscle biogenesis and lower muscle performance [88,89].

This study also showed that HITT- exercise for 12 weeks significantly increased lymphocyte apoptosis in healthy subjects. The data were significantly correlated with exercise duration, higher TAC and melatonin secretions, and a reduction or improvement of both adiposity and diabetes. The data obtained were in line with a previously estimated percentage of apoptotic lymphocytes isolated from the peripheral blood of normal subjects [90]. Furthermore, it was found previously that an exercise intensity threshold between 40 and 60% VO_2_ max significantly increases lymphocyte apoptosis [91].

The exercise showed efficiency in mobilizing and removing senescent T lymphocytes via apoptotic mechanisms from the bloodstream of healthy subjects following exercise training [92]. Apoptosis induced due to exercise may be caused by the interaction of death ligands and receptors such as the Fas receptor (CD95) and Fas ligand (CD95L), which were both significantly reported to increase following treadmill exercise tests [93]. The mobilized T lymphocytes from the peripheral lymphoid compartments were proposed to have a more advanced stage of biological aging with a reduction in capacity for clonal expansion than blood-resident T-cells, which ultimately were more sensitive to apoptosis than other blood lymphocytes [94].

Additionally, in this current study, there was a significant correlation between lymphocyte apoptosis and the duration of HIIT exercise training, which was significantly related to an increase in the level of melatonin with a prolonged time of daily exercise. This association supports the mechanistic role of melatonin in promoting lymphocyte apoptosis. Our data were in line with those who reported a significant role of melatonin in the stimulation of peripheral blood lymphocyte apoptosis, either via up-regulation of TNF, the major extrinsic mediator of apoptosis [94,95,96], or inhibition of lymphocyte division in the thymus and lymph nodes [96].

Our study had several limitations. Although our study generally showed the importance of HIIT exercise in enhancing physical performance among older adults and the significant association of lymphocyte apoptosis and melatonin as biological aging markers for aging changes among sedentary men aged 18–65 years, the lack of both a control group as well as long-term follow-up leads to a difficulty in seeing long-lasting changes in lymphocyte apoptosis and melatonin as a marker of biological aging. The lack of a control group is a major weakness. Our results can be interpreted as preliminary findings. Thus, further studies based on long follow-ups are recommended to understand the potential association of these biological parameters with phenotypic aging and to establish a future model of biological aging.

## 5. Conclusions

The main findings of this study were that HIIT exercise training interventions for 12 weeks significantly improved the adiposity markers, glycemic control parameters, and physical performance of sedentary, older adult men. In addition, melatonin secretion, % of lymphocyte apoptosis, COX activities, and TAC as biological aging markers were significantly increased following HIIT exercise training interventions for 12 weeks. The use of HIIT exercise seemed to be effective in improving biological aging, which will be adequate to support chronological age, especially regarding aging problems. However, subsequent studies are required with long-term follow up to consider HIIT as modulators for several cardiometabolic health problems in older individuals with obesity.

## Figures and Tables

**Figure 1 medicina-59-01201-f001:**
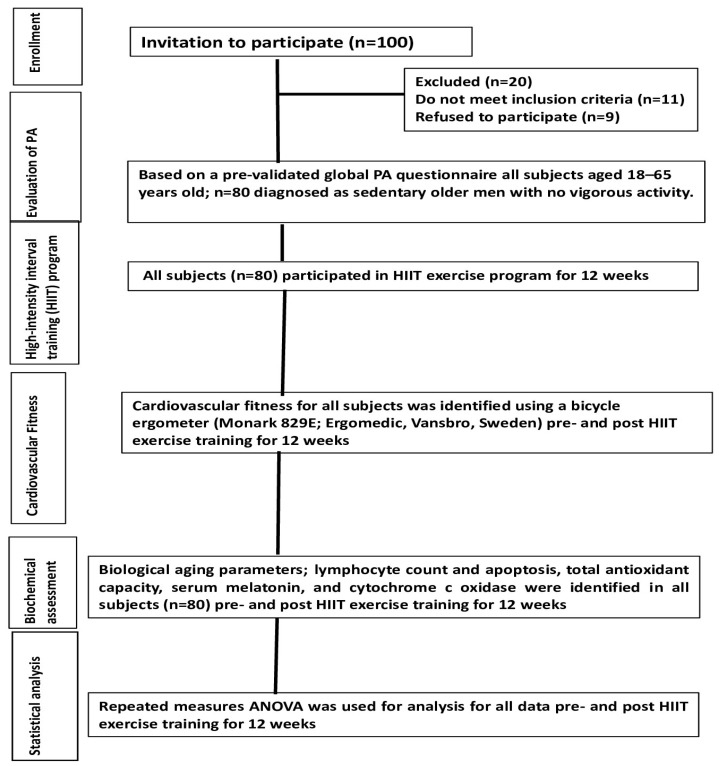
Outline of study procedures.

**Figure 2 medicina-59-01201-f002:**
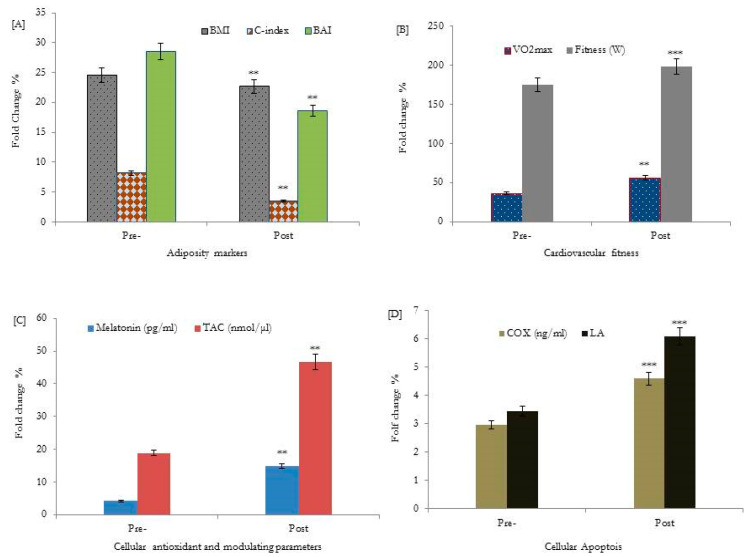
Effects of HIIT for 12 weeks on adiposity markers [**A**], cardiovascular fitness [**B**], cellular antioxidants and the melatonin modulating hormone [**C**], and cellular lymphocyte apoptosis [**D**] of healthy older adults. BMI: body mass index; WHtR: waist-to-height ratio; TAC: total antioxidant capacity; C-index: conicity index; BAI: body adiposity index; COX: cytochrome c oxidase (ng/mL). ** *p* < 0.01, bassline data of adiposity markers, VO_2_max, TAC, and cellular melatonin secretions were statistically significant compared to respective post-HIIT training values. *** *p* < 0.001, bassline data of fitness (W), and cellular apoptotic parameters; COX and LA % were statistically significant compared to respective values post-HIIT training intervention.

**Table 1 medicina-59-01201-t001:** Demographic and clinical characteristics of the participants (n = 80).

Parameters	Subjects (N = 80)
Mean age (years)	42.5 ± 3.1
BMI (kg/m^2^)	24.6 ± 2.3
Waist (cm)	92 ± 1.2
Hips (cm)	105 ± 0.65
WHR	0.87 ± 0.54
C-index	8.45 ± 0.24
BAI	29.1 ± 3.7
Fasting glucose (FG; mg/dL)	85.9 ± 7.3
Serum C-peptide (ng/mL)	3.95 ± 1.7
HbA1c (%)	4.7 ± 0.25
Fasting insulin (FI; μU/mL)	23.3 ± 3.1

Data expressed as mean ± SD, BMI: body mass index; WHR: waist to hip ratio, C-index: Conicity index; BAI: Body Adiposity Index; HbA1c (%):glycated hemoglobin.

**Table 2 medicina-59-01201-t002:** Changes in glycemic markers, BMI, WHtR, fitness scores, serum melatonin, cytochrome c, TAC, and lymphocyte apoptosis (%) in healthy subjects following 12 weeks of the HIIT exercise program.

Parameters	Subjects (n = 80)		
	Pre-	Post-	*p*-Value
BMI (kg/m^2^)	24.6 ± 2.3	22.7 ± 1.9 *	0.001
WHtR	0.87 ± 0.54	0.84 ± 0.58 *	0.001
C-index	8.45 ± 0.24	3.48 ± 0.12 **	0.001
BAI	29.1 ± 3.7	17.9 ± 2.7 **	0.001
Fitness (W)	178.4 ± 2.8	196.5 ± 4.6 **	0.001
VO_2_max (mL/kg/min)	48.2 ± 1.3	58.9 ± 2.5 **	0.001
Fasting glucose (mg/dL)	85.9 ± 7.3	82.3 ± 5.1 *	0.01
Serum C-peptide (ng/mL)	3.95 ± 1.7	4.1 ± 1.2 **	0.01
HbA1c (%)	4.7 ± 0.25	4.3 ± 0.65 **	0.01
Fasting insulin (FI; μU/mL)	23.3 ± 3.1	28.1 ± 2.7 **	0.001
Serum melatonin (pg/mL)	3.18 ± 0.98	11.2 ± 2.3 **	0.001
Cytochrome c oxidase (ng/mL)	2.8 ± 0.86	3.7 ± 0.75 **	0.001
TAC (nmol/μL)	21.5 ± 3.1	48.7 ± 7.1 **	0.002
% of Lymphocyte Apoptosis	3.15 ± 0.47	5.2 ± 0.31 **	0.003

Data expressed as mean ± SD, percentage (%), * *p* < 0.01. ** *p* < 0.001 (pre vs. post) level values of subjects pre- and post-exercise training. BMI: body mass index, WHtR: waist-to-height ratio, C-index: conicity index; BAI: body adiposity index; TAC: total antioxidant capacity.

**Table 3 medicina-59-01201-t003:** Correlation coefficients analysis of serum melatonin and lymphocyte apoptosis (%) as an antiaging hormone with adiposity markers, fitness, TAC, and cytochrome C in healthy subjects following 12 weeks of the HIIT- exercise program.

	Serum Melatonin				Lymphocyte Apoptosis (%)			
	β	r	OR (95% CI)	*p*-Value	β	r	OR (95% CI)	*p*-Value
**BMI** (kg/m^2^)	0.12	0.15	1.9 (1.3–2.6)	0.01	0.34	0.87	2.7 (1.5–3.1)	0.11
**WHtR**	0.21	0.31	2.4 (1.8–3.1)	0.01	0.25	0.36	1.2 (0.96–1.8)	0.14
**C-index**	0.24	0.48	1.9 (1.5–3.2)	0.01	0.31	0.47	1.5 (0.96–2.3)	0.05
**BAI**	0.19	0.21	3.1 (1.8–5.7)	0.001	0.27	0.41	1.7 (0.96–2.6)	0.01
**Fitness (W)**	0.54	0.23	4.5 (3.2–6.7)	0.001	0.15	0.87	10.2 (5.6–14.8)	0.001
**TAC** (nmol/µL)	0.45	0.68	6.4 (4.8–11.4)	0.001	0.45	0.89	5.1 (3.5–7.3)	0.001
**HbA1c (%)**	0.32	0.28	2.7 (1.8–3.8)	0.002	0.43	0.941	3.7 (2.1–4.8)	0.001
**Cytochrome C** (COX) (ng/mL)	0.76	0.47	5.1 (2.8–4.3)	0.001	0.381	0.518	7.9 (4.7–10.9)	0.001
**Exercise duration**	0.15	0.52	2.8 (1.5–3.8)	0.001	0.75	0.78	8.6 (5.1–10.4)	0.001
**Serum melatonin** (pg/mL)	-	-	-	-	0.48	0.37	2.9 (1.5–4.3)	0.002

Multinomial logistic regressions were used to estimate correlations of the studied variables. BMI: body mass index; WHtR: waist-to-height ratio, TAC: total antioxidant capacity; C-index: conicity index; BAI: body adiposity index; beta regression coefficient (β); correlation coefficient (r); ORs: odds ratios; CI: confidence interval.

## Data Availability

The dataset supporting the study findings has been presented in this study and will be available from the corresponding authors upon a reasonable request.

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
