# Peer review of "Effects of High-Intensity Interval Training on Melatonin Function and Cellular Lymphocyte Apoptosis in Sedentary Middle-Aged Men"

_medicina, 2023, doi:10.3390/medicina59071201_

Round 1

Reviewer 1 Report

This manuscript investigated the effect of high-intensity training on multiple markers of healthy aging. The paper is in general well-written, but following comments should be well-addressed:

Major:

1.      Please clarify how to generate table 3. What do “R” mean? It looks like you’re running logistic regression, but I have a hard time to find the description regarding how you set up the model. Could you please clarify and pinpoint where you described the details?

2.      It is very important to see how these changes truly link to the longitudinal changes in the functional level (aka, physical and cognitive functions). If you don’t have this information, at least you can cite previous publications that have showed the longitudinal changes in these markers have been linked to physical and cognitive functions. Indeed, studies have shown this to be true in a young-to-mid-age cohort (PMID: 33796868) and a cohort with age range extended to older adults (PMID: 36910594).

3.      It is very impressive that you see a change in an opposite direction of aging (PMID: 32107805). However, at two big questions should be mentioned in the discussion – (1) There is no control/placebo group. (2) There is no long-term follow-up, so it’s difficult to know whether this effect will last long.

Minor:

1.      In the method description, you probably don’t need the “()” when you talked about the “repeated measures (ANOVA)”. It can be straightforward repeated-measures ANOVA.

2.      If possible, put the unit of the variables in the table.

Author Response

I have attached a point-by-point response letter to Reviewer 1's comments. 

Kindly do the needful.

Reviewer 2 Report

General comments

The authors have clearly stated that the purpose of the study was to determine the effects of HIIT for 12 weeks on melatonin function, lymphocyte cell apoptosis, oxidative stress on aging, and physical performance. The paper is well-written, easy to follow and adds merit to the vital role of HIIT in health and performance for middle-aged men. Given this approach, this work can enhance future attempts in similar research area. However, I have highlighted a few suggestions and concerns in my specific comments section (below) that need to be addressed before considering whether this work should be published or not.

Specific comments

TITLE

Since the mean age of participants is 42.5 ± 3.1 years, the title should be slightly changed to the following option: “Effects of High-Intensity Interval Training on Melatonin Function and Cellular Lymphocyte Apoptosis in Sedentary Middle-Aged Men”.

ABSTRACT

- Exact percentage of change, effects sizes, and p values should be added in results regarding the primary outcome measures.

INTRODUCTION

- A statement about the popularity of HIIT in the global health and fitness industry should be added, according to report by the American College of Sports Medicine (1).

Suggested reference:

1.      Kercher VM, Kercher K, Levy P, Bennion T, Alexander C, Amaral PC, et al. 2023 Fitness Trends from Around the Globe. ACSMs Health Fit J 2023; 27(1): 19-30.

MATERIALS AND METHODS

-  Add the flowchart of the study as a Figure, showing also potential dropout.

- Dropout (if any) and attendance rates (% of prescribed exercise sessions) should be presented at the beginning of the results section.

-   Considering the dropout (if any), please clarify what kind of analysis was used (per protocol or intention-to-treat). In case of a dropout rate ≥20%, an intention-to-treat analysis is recommended given the small sample size. Please revise the results section, including tables and figures accordingly.

-    Clarify the exercise setting used for the experiment (lab or gym).

-    Clarify if the protocol was supervised (fully or partially) or not. If it was supervised, state who were in charge of the supervision.

-  Using the term “sedentary” in the title, you should describe the methodology used in order to make sure that all participants were sedentary or not. Did you used any relevant questionnaire (i.e., IPAQ), pedometer, or accelerometer? If so, please add this information in the methods section. If not, you have to remove the term “sedentary” or even “inactive”, since you did not assess participants’ habitual physical activity levels at baseline and throughout the 12-week intervention.

-     Did you controlled participants’ eating patterns through a diet recall? If so, add further details about that. If not, you have to add this particular point to the study limitations.

RESULTS

- Exact percentage of change, p-values, 95% confidence intervals, and effect sizes should be presented in results, including tables.

DISCUSSION

- Add the most recent evidence of the effectiveness of various exercise types, including HIIT on several cardiometabolic health-related parameters in overweight adults (2,3). The mean BMI and WHR of your sample were very close to the overweight category, which means that HIIT may be an effective exercise strategy for providing middle-aged men with a preventive effect on potential metabolic health impairments.

- Strengths and limitations should be stated in the same paragraph at the end of the discussion section. Lack of control group or a comparator (i.e., other training modality) should be mentioned. Also, habitual physical activity and eating habits assessments may be additional limitations (check a relevant comment above).

- In conclusions, you should underline the main findings and suggest future research attempts in this area while highlighting potential practical implications.

Suggested references:

2. https://pubmed.ncbi.nlm.nih.gov/35477256/

3. https://pubmed.ncbi.nlm.nih.gov/34679738/

Moderate editing of English language is required. There are several minor grammar and syntax errors as well as typos throughout the manuscript.

Author Response

I have attached a point-by-point response letter to Reviewer 2's comments. 

Kindly do the needful.

Round 2

Reviewer 1 Report

This manuscript has improved significantly. Some comments:

1. The authors should clearly mentioned that there is no control group in the study design, and clearly stated this in the limitation. This is especially important.

2. The authors should give some potential explanations regarding difference from published RCT results. (line 344-348) 

3. Furthermore, for the line 344-348, the cited paper 68 doesn't seem to be an RCT paper. Probably you should check your cited works.

4. Minor - Some typos are found - ex: "inturn" should be "in turn" in line . 312.  I have no issues following the English written here, but it's good to have a second look to fix some minor typos. 

Some typos are found - ex: "inturn" should be "in turn" in line . 312.  In general, I have no issues following the English written here. But it's good to have a second look to fix some minor typos. 

Author Response

Dear Concern

I have attached a copy of a point-by-point response letter to Reviewer 1's comments in round 2. 

Kindly do the needful.

Regards
